# Individual and contextual factors associated with abortion among reproductive age women in sub-Saharan Africa: Insights from 24 recent demographic and health surveys

**Beminate Lemma Seifu**[1]*, **Tsion Mulat Tebeje**[2], **Yordanos Sisay Asgedom**[3], **Zufan Alamrie Asmare**[4], **Hiwot Altaye Asebe**[5], **Bizunesh Fantahun Kase**[5], **Abdu Hailu Shibeshi**[6], **Afework Alemu Lombebo**[7], **Kebede Gemeda Sabo**[8], **Betel Zelalem Wubshet**[9], **Bezawit Melak Fente**[10], **Kusse Urmale Mare**[8]

1 Department of Public Health, College of Medicine and Health Sciences, Samara University, Samara, Ethiopia, 2 School of Public Health, College of health sciences and Medicine, Dilla University, Dilla, Ethiopia, 3 Department of Epidemiology and Biostatistics, College of Health Sciences and Medicine, Wolaita Sodo University, Wolaita Sodo, Ethiopia, 4 Department of Ophthalmology, School of Medicine and Health Science, Debre Tabor University, Debre Tabor, Ethiopia, 5 Department of Public Health, Collage of Medicine and Health Sciences, Samara University, Samara, Ethiopia, 6 Department of Statistics, College of Natural and Computational Science, Samara University, Samara, Ethiopia, 7 School of Medicine, College of Health Science and Medicine, Wolaita Sodo University, Wolaita Sodo, Ethiopia, 8 Department of Nursing, College of Medicine and Health Sciences, Samara University, Samara, Ethiopia, 9 Department of Nursing, College of Medicine and Health Sciences, Samara University, Afar, Ethiopia, 10 Department of Clinical Midwifery, School of Midwifery, College of Medicine & Health Sciences, University of Gondar, Gondar, Ethiopia

* beminetlemma1915@gmail.com

**Data Availability Statement:** Data is available online and can be accessed from https://

## Abstract

### Introduction

Despite the Sustainable Development Goal to reduce the global maternal mortality ratio to less than 70 per 100,000 live births by 2030, abortion remains one of the top five causes of maternal mortality in low and middle-income countries. However, there is a lack of comprehensive data on the pooled prevalence and determinants of abortion in sub-Saharan Africa (SSA). Therefore, this study aims to investigate the pooled prevalence and determinants of abortion among women of reproductive age in 24 SSA countries using the most recent Demographic and Health Surveys.

### Methods

The most recent Demographic and Health Survey (DHS) data from 24 Sub-Saharan African (SSA) countries were analyzed, using a weighted sample of 392,332 women of reproductive age. To address the clustering effects inherent in DHS data and the binary nature of the outcome variable, a multilevel binary logistic regression model was employed. The results were reported as adjusted odds ratios with 95% confidence intervals to indicate statistical significance. Additionally, the model with the lowest deviance was identified as the best fit for the data.

dhsprogram.com/data/dataset_admin/index.cfm.
We have also uploaded the dataset as supporting
information

**Funding:** The author(s) received no specific
funding for this work.

**Competing interests:** The authors have declared
that no competing interests exist.

**Abbreviations:** AIC, Akaike Information Criteria;
AOR, Adjusted Odds Ratio; BIC, Bayesian
Information Criteria; CI, Confidence Interval; DHS,
Demographic health survey; EAs, Enumeration
areas; LMICs, Lower and Middle-income Countries;
LLR, Log likelihood ratio; LR, Likelihood ratio;
RAW, Reproductive Age Women; SSA, Sub-
Saharan Africa; WHO, World Health Organizations.

## Results

The pooled prevalence of abortion in SSA were 6.93% (95%CI: 5.38, 8.48). Older age (AOR = 3.71; 95%CI: 3.46, 3.98), ever married (AOR = 3.87; 95%CI: 3.66, 4.10), being educated (AOR = 1.35; 95%CI: 1.28, 1.44), having formal employment (AOR = 1.19; 95%CI: 1.16, 1.23), traditional contraceptive use (AOR = 1.27; 95%CI: 1.19, 1.36) and media exposure (AOR = 1.37; 95%CI: 1.32, 1.41) found to be a predisposing factors for abortion. While high parity (AOR = 0.72; 95%CI: 0.68, 0.76), rural residence (AOR = 0.87; 95%CI: 0.85, 0.91), and rich (AOR = 0.96; 95%CI: 0.93, 0.99) wealth index were a protective factors.

## Conclusion

The study found that the pooled prevalence of abortion in Sub-Saharan Africa is 7%. Potential interventions include comprehensive sexual education to inform and empower women, increased access to modern contraceptives to reduce unintended pregnancies, improved healthcare services especially in rural areas, economic empowerment through education and employment opportunities, media campaigns to disseminate information and reduce stigma, and policy development to ensure safe and legal access to abortion services. These interventions aim to improve reproductive health outcomes and reduce unsafe abortions in SSA.

## Introduction

According to the World Health Organization (WHO), abortion is defined as the termination of pregnancy before 20 weeks of gestational age [1]. Abortion can occur either through medical intervention, such as medication or surgical procedures, or naturally, as in the case of a miscarriage [2]. Annually, more than 73 million abortions are performed globally. Six out of ten (61%) unwanted pregnancies and three out of ten (29%) of all pregnancies end in induced abortion [3]. In Africa, an estimated 33 abortions are performed per 1,000 reproductive age women each year, with minimal difference throughout Eastern, Middle, Southern, and Western Africa. This rate has stayed stable over the last two decades [4]. However, due to population expansion, the annual number of abortions in Sub-Saharan Africa doubled between 1995–1999 and 2015–2019, rising from 4.3 million to 8.0 million [5]. The doubling of annual abortions in Sub-Saharan Africa (SSA) can be attributed to several factors. Limited access to modern contraceptives leads to higher rates of unintended pregnancies. Socioeconomic challenges, such as poverty and lack of education, also play a significant role. Additionally, restrictive abortion laws and inadequate healthcare services force many women to seek unsafe abortion methods, further contributing to the increase. Addressing these issues through improved access to contraception, education, and healthcare services is crucial to mitigating the rise in abortions [4].

Globally 45% of all induced abortions are unsafe. Around One-third of all unsafe abortions in Africa performed under the most hazardous states, i.e. by unskilled individuals utilizing injurious and infectious techniques [1]. Despite the Sustainable Development Goal to reduce the global maternal mortality ratio to less than 70 per 100,000 live births by 2030, abortion remains a significant cause of maternal deaths, accounting for 37 deaths per 100,000 live births in SSA and 12 per 100,000 in South Asia. Maternal death related to abortion, particularly the unsafe abortion, contributes to 13% of maternal deaths worldwide [6].

According to previous studies done in different countries abortion were found to be associated with age [7,8], educational status [9,10], residence [11,12], wealth status [13], marital status [14], employment [15], parity [7], woman's nutritional status [16,17], substance use, preceding birth interval [18,19], ANC visit, contraceptive use [7], Partner's educational status, and media exposure [15].

Even though abortion is undeniably associated with pregnancy and childbirth complications, and it poses a significant threat to maternal health and mortality rates, especially in countries with low and middle incomes, currently there is a lack of comprehensive data on the pooled prevalence and contributing factors of abortion in sub-Saharan Africa countries. Previous studies on abortion in SSA have often been limited in scope, focusing on specific regions [15,20,21] or particular age groups, such as adolescents [22]. This has resulted in a fragmented understanding of the issue, highlighting the need for more comprehensive research that encompasses a broader demographic and geographic range. This study aims to fill that gap by examining the pooled prevalence and determinants of abortion among women of reproductive age across 24 SSA countries using the most recent Demographic and Health Surveys.

**Hypothesis:** *Individual factors such as age, marital status, educational status, partner's/husband's educational status, occupation, wealth index, parity, contraceptive use, ANC visit, preceding birth interval, and media exposure., along with contextual factors like access to healthcare services, place of residence significantly influence the odds of abortion among reproductive-age women in Sub-Saharan Africa.*

## Methods

### Data source, study setting and population

The most recent Demographic and Health Survey (DHS) data from 24 SSA nations were used in this study. DHS is a nationally representative survey routinely conducted every five years and gathers data regarding basic health parameters such as mortality, morbidity, fertility, and maternal and child health-related characteristics. The survey used a two-stage stratified sampling technique to select the study participants. In the first stage, Enumeration Areas (EAs) were randomly selected based on the country's recent population and using the housing census as a sampling frame, households were randomly selected in the second stage. Men, women, children, birth, and household datasets are all included in each country's survey. Because, the study population was reproductive-age women, we used the individual (women's) Record dataset (IR file). In the current study, 392,332 women of reproductive age were considered for final analysis. Detailed information about DHS methodology can be found from the official database https://dhsprogram.com/Methodology/index.cfm.

### Study variables

**Dependent variable:** This study's outcome variable was abortion among RAW, this was gathered from the DHS inquiry "have you ever had a terminated pregnancy?" and divided into two categories: "Yes" if the woman has an abortion, whether spontaneous or induced (pregnancy termination before seven months), or "No" if she had not.

**Independent variables:** The individual and community level explanatory variables were selected based on their association with the outcome variable reported in previous studies and literatures, as well as their availability in the DHS datasets.

**The Individual-level variables:** age, marital status, educational status, partner's/husband's educational status, occupation, wealth index, parity, contraceptive use, ANC visit, preceding birth interval, and media exposure.

**The community-level variables:** residence, distance from health facility and sub-Saharan Africa regions.

## Operational definition

**Wealth Index**: in the Demographic and Health Surveys (DHS) is a composite measure of a household's cumulative living standard. It is calculated using data on a household's ownership of selected assets, such as televisions and bicycles; materials used for housing construction; and types of water access and sanitation facilities. Households are then categorized into five wealth quintiles (poorest, poorer, middle, richer, and richest) based on their relative wealth [23].

**Media exposure**: was created from three variables (frequency of listening to the radio, watching television, and reading newspapers or magazines). In this study, women who listened to radio or watched television or read newspaper/magazine at least less than once a week were considered as having exposure to media (coded "Yes") and otherwise labeled as not having media exposure (coded "No").

**Distance to a Health Facility:** This is typically measured based on the respondent's self-reported travel time or distance to the nearest health facility. It is classified as either a significant problem or not a significant problem. This measure includes:

Travel Time: The time it takes for an individual to reach the nearest health facility, often categorized into intervals (e.g., less than 30 minutes, 30–60 minutes, more than 60 minutes).

Physical Distance: The actual distance to the nearest health facility, this can be measured in kilometers or miles [24].

## Data management and analysis

Data of 24 Sub-Saharan African countries were pooled, recoded and analyzed using Stata version 17 software. Before analysis, each countries dataset were appended to create a single dataset. Appending is used when we want to combine datasets that contain the same variables, but have different cases, thus we are adding new rows to the dataset, but the number of columns will remain the same. This was achieved using the "***append using***" command in STATA.

Data were weighted using sampling weight, primary sampling unit, and strata to restore the survey's representativeness and obtain appropriate estimate. Descriptive results were presented using weighted frequencies and percentages.

To account for the clustering effects of DHS data and the binary nature of the outcome variable, a multilevel binary logistic regression model was applied to determine the effects of each independent variable on the outcome variable. Bivariable multilevel binary logistic regression analysis done to identify variables eligible for the multivariable analysis. Variables with a p-value less than 0.20 in this analysis and those found important in the literature were considered as candidates for multivariable multilevel binary logistic regression analysis. The choice of a p-value < 0.2 for selecting variables in the bi-variable analysis is intentional to ensure that potentially relevant variables are not excluded too early in the analysis process. This threshold is higher than the conventional 0.05 used for statistical significance, allowing us to capture a broader range of variables that may have an impact. By doing so, we aim to include variables that might show significance in the multivariable analysis, thus providing a more comprehensive understanding of the factors at play.

Four models were constructed for the multilevel binary logistic regression. The first model was a null model without explanatory variables to determine the extent of cluster variation in abortion. The second model was fitted with individual-level variables, the third with community-level variables, and the fourth with both individual and community-level variables at the same time. Variables with a p-value < 0.2 in the bi- variable multilevel binary logistic

regression analysis were considered for the multivariable analysis. Deviance was used to verify model fitness and a model with the lowest deviance was considered the best-fit model. Deviance is often preferred over AIC (Akaike Information Criteria) and BIC (Bayesian Information Criteria) for assessing the goodness-of-fit in nested models because it allows for a direct comparison using the likelihood ratio test, which provides a p-value to determine statistical significance. This focus on model fit, without the complexity penalties inherent in AIC and BIC, makes deviance simpler and more interpretable for nested models. While AIC and BIC are useful for model selection, especially with non-nested models, deviance offers a clearer measure of improvement in fit between nested models [25,26].

Finally, the Adjusted Odds Ratio (AOR) along with its 95% confidence interval (CI) was presented, highlighting variables that had a p-value of less than 0.05 in the multivariable analysis.

### Ethical consideration

This study did not require ethical approval or participant consent because it was a secondary data analysis of publicly available survey data from the MEASURE DHS program. We have obtained permission to download and use the data from http://www.dhsprogram.com for this study. There are no names or addresses of individuals or households recorded in the datasets.

## Result

### Background characteristics and prevalence of abortion among respondents

In this study, 392,332 reproductive age women were included. Of those 235,203 (59.95%) of them were from rural residencies. Majority (173,154 (44.13%)) of the women were from western Africa region, while 116,182 (29.61%), 46,282 (11.80%) and 56,714 (14.46%) of them were from Eastern, Central and Southern Africa regions respectively. Women who have history of terminated pregnancy were 50,809 (12.95%). More than half (61.89%) of RAW included in this study were married (Table 1).

The prevalence of abortion among women whose age were between 15–19 were 1.77% and 6.28%, 8.77%, 9.47% and 9.45% among whose age were 20–24, 25–29, 30–34 and ≥35 respectively. Abortion was prevalent in 54.98% of women who had previously terminated pregnancies. Among urban residents, the prevalence of abortion was 8.36% and 6.29% among rural residents. Furthermore, the prevalence of abortion was 12.47% among traditional contraceptive users and 9.01% among Primiparous mothers (Table 1).

### The pooled prevalence of abortion among reproductive age women in Sub-Sahara African countries

The pooled prevalence of abortion among RAW in SSA were 6.93% (95%CI: 5.38, 8.48). The prevalence of abortion in SSA ranges from 4.88% (95%CI: 4.57, 5.20) in Central Africa to 8.43% (95%CI: 5.59, 11.27) in West Africa (Fig 1).

### Statistical analysis and model comparison

Even though, the ICC value was less than 10% the Log-likelihood Ratio (LR) was significant, indicating that a multilevel binary logistic regression model better fits the data than the classical regressions. The Log-likelihood ratio test which was ($X^2$ = 85.15, $p$-value < 0.001) informed us to choose the generalized linear mixed-effect model (GLMM) over the basic model. The models were compared with deviance and the final model with both individual

**Table 1. Socio-demographic characteristics and the prevalence of abortion among reproductive age women in sub-Saharan Africa.**

| Variable | Total weighted frequency (%) | Abortion | |
|---|---|---|---|
| | | No (%) | Yes (%) |
| **Individual level variables** | | | |
| **Maternal age** | | | |
| 15–19 | 83,394 (21.26) | 81,914 (98.23) | 1,480 (1.77) |
| 20–24 | 72,413 (18.46) | 67,868 (93.72) | 4,544 (6.28) |
| 25–29 | 66,127 (16.85) | 60,330 (91.23) | 5,797 (8.77) |
| 30–34 | 55,658 (14.19) | 50,385 (90.53) | 5,273 (9.47) |
| ≥35 | 114,740 (29.25) | 103,900 (90.55) | 10,840 (9.45) |
| **Marital status** | | | |
| Never in union | 113,958 (29.05) | 111,399 (97.75) | 2,559 (2.25) |
| Married | 242,831 (61.89) | 220,395 (90.76) | 22,436 (9.24) |
| Widowed/ divorced/ separated | 35,543 (9.06) | 32,604 (91.73) | 2,939 (8.27) |
| **Maternal education** | | | |
| No formal education | 108,382 (27.63) | 101,002 (93.19) | 7,380 (6.81) |
| Primary | 126,196 (32.17) | 117,492 (93.10) | 8,704 (6.90) |
| Secondary | 132,522 (33.78) | 123,161 (92.94) | 9,361 (7.06) |
| Higher | 25,232 (6.43) | 22,744 (90.14) | 2,488 (9.86) |
| **Maternal employment** | | | |
| Not employed | 164,752 (41.99) | 156,186 (94.80) | 8,566 (5.20) |
| Employed | 227,580 (58.01) | 208,212 (91.49) | 19,368 (8.51) |
| **Wealth index** | | | |
| Poor | 139,852 (35.65) | 131,238 (93.84) | 8,614 (6.16) |
| Middle | 75,412 (19.22) | 70,238 (93.14) | 5,174 (6.86) |
| Rich | 177,068 (45.13) | 162,921 (92.01) | 14,147 (7.99) |
| **Partner's education (n = 242,570)** | | | |
| No formal education | 84,613 (34.88) | 78,317 (92.56) | 6,296 (7.44) |
| Primary | 69,524 (28.66) | 63,705 (91.63) | 5,819 (8.37) |
| Secondary | 66,271 (27.32) | 58,983 (89.00) | 7,287 (11.00) |
| Higher | 22,162 (9.14) | 19,154 (86.42) | 3,008 (13.58) |
| **History of terminated pregnancy** | | | |
| No | 341,523 (87.05) | 341,523 (100.00) | 0 (0.00) |
| Yes | 50,809 (12.95) | 22,875 (45.02) | 27,934 (54.98) |
| **Parity** | | | |
| Nulliparous | 110,470 (28.16) | 106,851 (96.72) | 3,619 (3.28) |
| Primiparous | 57,205 (14.58) | 52,051 (90.99) | 5,154 (9.01) |
| multiparous | 137,393 (35.02) | 124,868 (90.88) | 12,523 (9.12) |
| Grand multiparous | 87,263 (22.24) | 80,626 (92.39) | 6,637 (7.61) |
| **Contraceptive method** | | | |
| Non-user | 82,563 (72.02) | 263,599 (93.29) | 18,964 (6.71) |
| Traditional | 10,474 (2.67) | 9,168 (87.53) | 1,306 (12.47) |
| Modern | 99,295 (25.31) | 91,630 (92.28) | 7,665 (7.72) |
| **Preceding birth interval (n = 223,897)** | | | |
| < 24 | 35,829 (16.00) | 33,142 (92.50) | 2,687 (7.50) |
| ≥ 24 | 188,068 (84.00) | 171,671 (91.28) | 16,397 (8.72) |
| **ANC visit (n = 182,858)** | | | |
| None | 20,342 (11.12) | 19,399 (95.37) | 943 (4.63) |

(*Continued*)

**Table 1.** (Continued)

| Variable | Total weighted frequency (%) | Abortion | |
|---|---|---|---|
| | | No (%) | Yes (%) |
| 1–3 | 54,986 (54,986) | 51,699 (94.02) | 3,287 (5.98) |
| ≥ 4 | 107,530 (58.81) | 98,589 (91.68) | 8,942 (8.32) |
| **Media exposure** | | | |
| No | 117,250 (29.89) | 111,346 (94.96) | 5,904 (5.04) |
| Yes | 275,071 (70.11) | 253,040 (91.99) | 22,031 (8.01) |
| **Community level variables** | | | |
| **Residence** | | | |
| Urban | 157,129 (40.05) | 143,986 (91.64) | 13,143 (8.36) |
| Rural | 235,203 (59.95) | 220,411 (93.71) | 14,792 (6.29) |
| **Distance to the health facility** | | | |
| Big problem | 128,728 (34.55) | 120,444 (93.56) | 8,284 (6.44) |
| Not a big problem | 243,849 (65.45) | 225,832 (92.61) | 18,017 (7.39) |
| **Sub-Saharan Africa region** | | | |
| Eastern Africa | 116,182 (29.61) | 107,523 (92.55) | 8,659 (7.45) |
| Southern Africa | 56,714 (14.46) | 54,670 (96.40) | 2,044 (3.60) |
| Western Africa | 173,154 (44.13) | 158,353 (91.45) | 14,801 (8.55) |
| Central Africa | 46,282 (11.80) | 43,851 (94.75) | 2,431 (5.25) |

and community level variables was chosen as the best-fitted model since it had lowest deviance value (186,641) (Table 2).

## Factors associated with abortion among reproductive age women in Sub-Saharan Africa

In the final multivariable multilevel binary logistic regression model: age, marital status, maternal education, maternal employment, wealth index, parity, contraceptive use, media exposure, residence and SSA region were found to be statistically significant ($p\ value$ <0.05) determinants of abortion.

The likelihood of having history of abortion among rural residents were nearly 13% (AOR = 0.87; 95%CI: 0.85, 0.91) lower compared to women who reside in urban. The odds of experiencing abortion among married and widowed/divorced/separated were 3.87 times (AOR = 3.87; 95%CI: 3.66, 4.10) and 3.02 times (AOR = 3.02; 95%CI: 2.82, 3.23) higher compared to never married women. Women who had a formal employment were 19% (AOR = 1.19; 95%CI: 1.16, 1.23) more likely to experience abortion compared to women who do not have a formal employment. Regarding contraceptive method used, the odds of abortion among women who use traditional methods were 27% (AOR = 1.27; 95%CI: 1.19, 1.36) higher than women who do not use any methods while the likelihood of experiencing abortion were by 7% (AOR = 0.93; 95%CI: 0.91, 0.96) lower among modern contraceptive users compared to women who do not use any contraceptive methods. Having media exposure were associated with a 37% (AOR = 1.37; 95%CI: 1.32, 1.41) higher odds of abortion (Table 3).

## Discussion

Our pooled data from this cross-sectional study of 392,332 reproductive-age women from 24 SSA nations revealed that the weighted pooled prevalence of abortion was 6.93%, with a considerable variation between countries. Age, marital status, maternal education, maternal

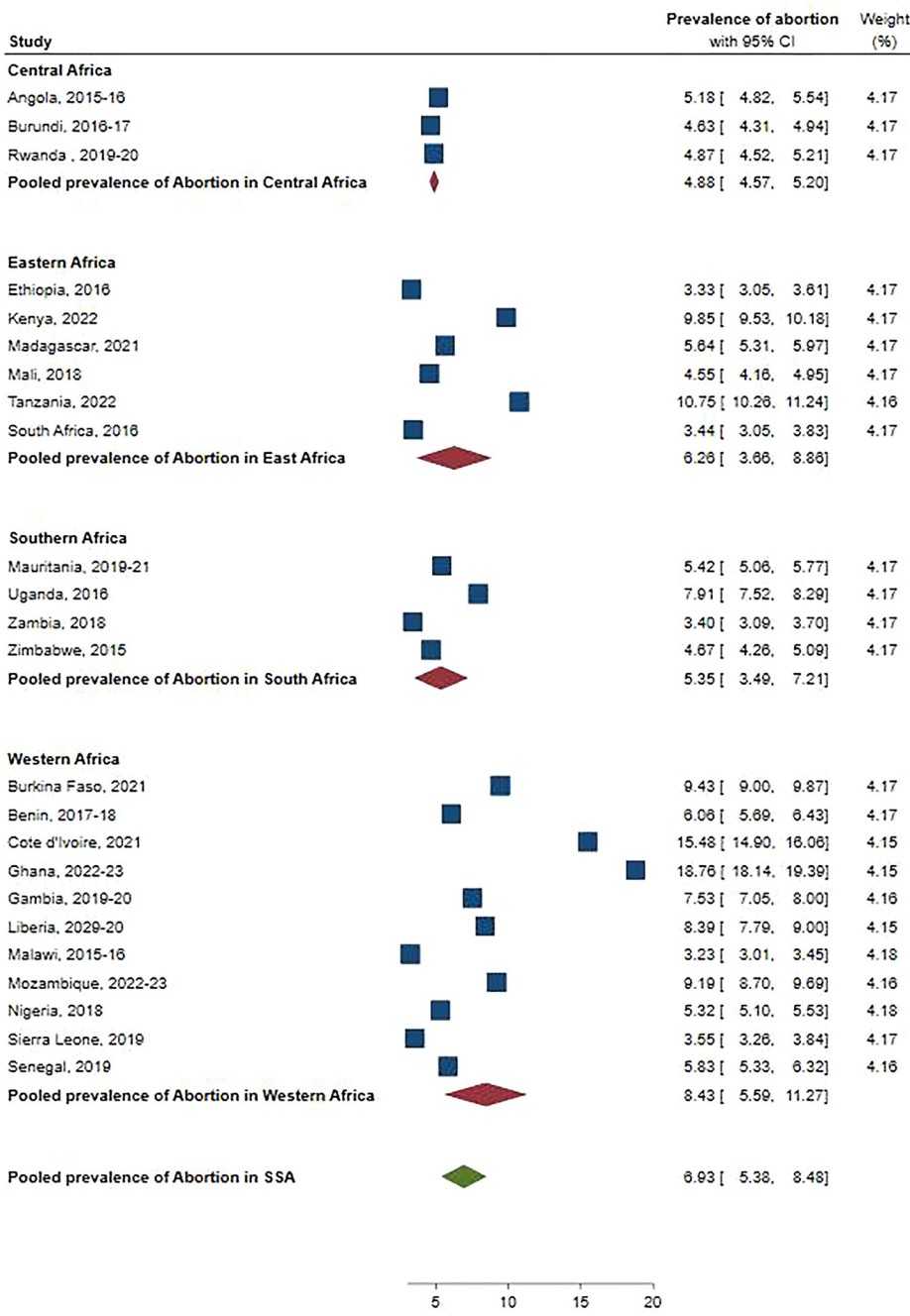

**Fig 1. The pooled prevalence of abortion among reproductive age women in SSA regions and SSA.**

employment, wealth index, parity, contraceptive use, media exposure, residence and SSA region were significantly associated with abortion.

The pooled prevalence of abortion among RAW in SSA were 6.93% (95%CI: 5.38, 8.48). The high prevalence of abortion in SSA could be attributed to the region's highest pregnancy rate (218 per 1,000 women per year) as well as the highest rate of unwanted pregnancy (91 per 1,000). In SSA, an estimated 37% of women who become pregnant unintentionally terminate

**Table 2. Model comparison and random effect results.**

|  | Null model | Model 1 | Model 2 | Model 3 |
|---|---|---|---|---|
| Log likelihood | -98173.64 | - 93753 | -97634.8 | 93307.5 |
| Deviance | 196,347.28 | 187506 | 195,269.6 | 186615 |
| AIC | 196351.3 | 187546 | 195281.6 | 186663 |
| BIC | 196373.3 | 187766.6 | 195347.8 | 186927.7 |
| LR test | X$^2$ = 85.15, $p$-value < 0.001) |  |  |  |

LR: Log-likelihood ratio test, AIC: Akaike's information criterion, BIC: Bayesian information criterion.

the pregnancy [4]. Furthermore a low prevalence of contraceptive use in SSA is another contributing factor for the high prevalence of abortion [27]. The prevalence of abortion in Western Africa was 8.43% (95%CI: 5.59, 11.27), which surpasses that of other regions. The higher prevalence of abortion in Western Africa can be attributed to several factors, including strong cultural and religious stigmas that lead to unsafe practices, limited access to education and reproductive health information, and economic challenges such as poverty and urbanization pressures. Additionally, restrictive abortion laws and inadequate healthcare infrastructure contribute to higher rates of unsafe abortions [28–30]. These socio-cultural, economic, and legal factors collectively influence the higher abortion prevalence in the region.

This study showed that RAW whose age is 20 and above were more likely to experience abortion compared to youth whose age is between 15–19 years, this is in line with studies done in united states of America (USA) [31], Denmark [32] and 27 SSA countries [33]. Women of older ages have a greater likelihood to have a high-risk pregnancy, which might result in abortion due to maternal medical conditions and conception-related diseases such as preeclampsia, ectopic pregnancy, and gestational diabetes [34,35].

Mothers with a larger (5 or more) number of children were less likely to experience abortion. This corroborated by findings from previous studies [36,37]. This may be explained by the fact that moms with greater parity may have more understanding of menstrual cycles and use of maternal health services like family planning. These mothers may also be aware that using contraception is the most effective way to restrict the number of children and increase birth spacing.

The odds of experiencing abortion among ever-married women were high, compared to women who has never been in union. The possible explanation for the positive association between marital status and abortion could be the lack of contraceptive use or contraceptive failure [38] besides studies reported that married women use abortion as a means of birth spacing and limiting to attain their child bearing goals [39].

Backed by earlier researches [40–43], this study shows that women with a higher educational status, wealth index, and formal employment were more likely to have abortions than their counterparts. A reasonable explanation could be that educated women, particularly in low and middle-income nations like SSA, may have pregnancies that interfere with their education which predispose them to elect to terminate those pregnancies. Furthermore, formally employed and women who have a high wealth status may view pregnancy as a hindrance to their careers and pursuit of high productivity and revenue. As a result, if they believe they have had their desired number of children, they do not hesitate to abort an unwanted pregnancy [44].

Our study corroborates the findings from previous studies, which have reported women who have had media exposure were more likely to have abortion than those who had not [37,45]. In low and middle income countries like SSA media exposure is related to self-efficacy

**Table 3. A multilevel analysis of determinants of abortion among reproductive age women in sub-Saharan Africa.**

| Variable | Model 1 AOR (95%CI) | Model 2 AOR (95%CI) | Model 3 AOR (95%CI) |
|---|---|---|---|
| **Individual level variables** | | | |
| **Maternal age** | | | |
| 15–19 | 1 | | 1 |
| 20–24 | 2.24 (2.10, 2.39) | | 2.27 (2.13, 2.43)* |
| 25–29 | 2.98 (2.78, 3.18) | | 3.01 (2.81, 3.22)* |
| 30–34 | 3.32 (3.09, 3.56) | | 3.41 (3.18, 3.66)* |
| ≥35 | 3.61 (3.36, 3.86) | | 3.71 (3.46, 3.98)* |
| **Marital status** | | | |
| Never in union | 1 | | 1 |
| Married | 4.04 (3.82, 4.28) | | 3.87 (3.66, 4.10)* |
| Widowed/ divorced/ separated | 3.03 (2.83, 3.24) | | 3.02 (2.82, 3.23)* |
| **Maternal education** | | | |
| No formal education | 1 | | 1 |
| Primary | 1.12 (1.09, 1.16) | | 1.33 (1.28, 1.38)* |
| Secondary | 1.21 (1.17, 1.26) | | 1.37 (1.32, 1.43)* |
| Higher | 1.24 (1.17, 1.31) | | 1.35 (1.28, 1.44)* |
| **Maternal employment** | | | |
| Not employed | 1 | | 1 |
| Employed | 1.22 (1.19, 1.26) | | 1.19 (1.16, 1.23)* |
| **Wealth index** | | | |
| Poor | 1 | | 1 |
| Middle | 1.02 (0.99, 1.06) | | 0.99 (0. 95, 1.03) |
| Rich | 1.04 (1.01, 1.07) | | 0.96 (0.93, 0.99)* |
| **Parity** | | | |
| Nulliparous | 1 | | 1 |
| Primiparous | 1.02 (0.97, 1.07) | | 1.04 (0.99, 1.10) |
| multiparous | 0.70 (0.66, 0.74) | | 0.72 (0.68, 0.76)* |
| Grand multiparous | 0.55 (0.52, 0.58) | | 0.56 (0.53, 0.60)* |
| **Contraceptive method** | | | |
| Non-user | 1 | | 1 |
| Traditional | 1.29 (1.21, 1.38) | | 1.27 (1.19, 1.36)* |
| Modern | 0.84 (0.82, 0.87) | | 0.93 (0.91, 0.96)* |
| **Media exposure** | | | |
| No | 1 | | 1 |
| Yes | 1.45 (1.40, 1.50) | | 1.37 (1.32, 1.41)* |
| **Community level variables** | | | |
| **Residence** | | | |
| Urban | 1 | 1 | 1 |
| Rural | | 0.81 (0.79, 0.84) | 0.87 (0.85, 0.91)* |
| **Sub-Saharan Africa region** | | | |
| Eastern Africa | 1 | 1 | 1 |
| Southern Africa | | 0.62 (0.59, 0.66) | 0.68 (0.65, 0.72)* |
| Western Africa | | 0.45 (0.43, 0.48) | 0.45 (0.43, 0.48)* |
| Central Africa | | 1.09 (1.06, 1.12) | 1.14 (1.11, 1.17)* |

in abortion decision-making among adolescent girls and young women [45]. Furthermore, the internet and other channels that have interpenetrate most young people's lives might be utilized to find information regarding all the places and methods that women could use to abort a pregnancy.

While using modern contraceptive use were negatively associated with abortion, traditional contraceptive use were positively associated. This is supported by previous studies [15,46]. This finding may have been justified by the fact that modern contraception users had a lower likelihood of unwanted/unplanned pregnancy than traditional users, that is frequently ended before the fetus reaches the age of viability [46,47].

The odds of abortion found to be varied across sub-Saharan Africa regions and place of residence; this could be due to the variation of availability and accessibility of family planning service [48], magnitude of unintended pregnancy [49] and maternal health care services [50].

## Strength and limitation of the study

This research is one of the few that examine the pooled prevalence and determinants abortion using the latest DHS data available from several SSA countries with a large sample size, that were representative of the national population. Furthermore, this study used a weighted dataset with powerful statistical analytic techniques, which attribute the correlated nature of the DHS data and provides us with reliable estimates and standard errors. This study, however, is not without limitations. Because the DHSs are cross-sectional, we cannot prove the causal relationship between the independent variables and abortion. Furthermore, because the data gathered through interviews, there is a risk of recall bias, and this study does not distinguish between spontaneous and induced abortion.

## Implication of the study

Studying the pooled prevalence of abortion among women of reproductive age in Sub-Saharan Africa (SSA) is crucial for public health planning, policy development, and resource allocation. It helps address stigma, reduce maternal mortality, and improve reproductive health outcomes by providing insights into the scale and distribution of abortion, guiding the development of supportive policies, and informing educational programs.

## Conclusion

The study indicates a high prevalence of abortion in SSA, highlighting its significance as a public health issue. Older age, ever married, being educated, having formal employment, traditional contraceptive use and media exposure found to be a predisposing factors for abortion. While high parity, rural residence, middle and rich wealth index were a protective factors. To reduce the prevalence of abortion and improve overall public health in SSA, it is essential to implement comprehensive sexual and reproductive health education programs to increase awareness about safe contraceptive methods and the risks associated with abortion. Ensuring the availability and accessibility of modern contraceptives, especially in rural areas can help reduce reliance on traditional methods. Additionally, enhancing the capacity of healthcare facilities to provide safe abortion services and post-abortion care, particularly in underserved regions, is crucial. Addressing socioeconomic disparities by creating employment opportunities and improving living conditions can indirectly reduce the incidence of abortion. Furthermore, utilizing media platforms to disseminate accurate information about reproductive health and the importance of using effective contraceptive methods is vital. By addressing these areas, we can work towards reducing the prevalence of abortion and improving overall public health in SSA.

## Supporting information

**S1 Dataset.**
(RAR)

## Acknowledgments

We would like to thank the measure DHS program for providing the datasets.

## Author Contributions

**Conceptualization:** Beminate Lemma Seifu, Zufan Alamrie Asmare, Hiwot Altaye Asebe, Bizunesh Fantahun Kase, Abdu Hailu Shibeshi, Afework Alemu Lombebo, Kebede Gemeda Sabo, Betel Zelalem Wubshet, Bezawit Melak Fente, Kusse Urmale Mare.

**Data curation:** Beminate Lemma Seifu, Yordanos Sisay Asgedom, Zufan Alamrie Asmare, Hiwot Altaye Asebe, Bizunesh Fantahun Kase, Abdu Hailu Shibeshi, Afework Alemu Lombebo, Kebede Gemeda Sabo, Betel Zelalem Wubshet, Bezawit Melak Fente, Kusse Urmale Mare.

**Formal analysis:** Beminate Lemma Seifu, Tsion Mulat Tebeje, Yordanos Sisay Asgedom, Zufan Alamrie Asmare, Abdu Hailu Shibeshi, Afework Alemu Lombebo, Kebede Gemeda Sabo, Bezawit Melak Fente, Kusse Urmale Mare.

**Methodology:** Beminate Lemma Seifu, Hiwot Altaye Asebe, Bizunesh Fantahun Kase, Afework Alemu Lombebo, Betel Zelalem Wubshet, Bezawit Melak Fente, Kusse Urmale Mare.

**Software:** Beminate Lemma Seifu, Tsion Mulat Tebeje, Zufan Alamrie Asmare, Kusse Urmale Mare.

**Supervision:** Beminate Lemma Seifu, Kusse Urmale Mare.

**Visualization:** Bizunesh Fantahun Kase, Kebede Gemeda Sabo.

**Writing – original draft:** Beminate Lemma Seifu, Tsion Mulat Tebeje, Yordanos Sisay Asgedom, Zufan Alamrie Asmare, Hiwot Altaye Asebe, Bizunesh Fantahun Kase, Abdu Hailu Shibeshi, Afework Alemu Lombebo, Kebede Gemeda Sabo, Betel Zelalem Wubshet, Bezawit Melak Fente, Kusse Urmale Mare.

**Writing – review & editing:** Beminate Lemma Seifu, Tsion Mulat Tebeje, Yordanos Sisay Asgedom, Zufan Alamrie Asmare, Hiwot Altaye Asebe, Bizunesh Fantahun Kase, Abdu Hailu Shibeshi, Afework Alemu Lombebo, Kebede Gemeda Sabo, Betel Zelalem Wubshet, Bezawit Melak Fente, Kusse Urmale Mare.

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
