## [Decision Letter · Decision Letter 0]

21 Aug 2024

PONE-D-23-16508Individual and contextual factors associated with abortion among reproductive age women in Sub-Saharan Africa: a generalized linear mixed effect modelingPLOS ONE

Dear Dr. Seifu,

Thank you for submitting your manuscript to PLOS ONE. After careful consideration, we feel that it has merit but does not fully meet PLOS ONE’s publication criteria as it currently stands. Therefore, we invite you to submit a revised version of the manuscript that addresses the points raised during the review process.

We look forward to receiving your revised manuscript.

Kind regards,

Akaninyene Eseme Bernard Ubom, MBBS, MWACS, OMI Fellow

Academic Editor

PLOS ONE

Reviewers' comments:

Reviewer's Responses to Questions

**Comments to the Author**

1. Is the manuscript technically sound, and do the data support the conclusions?

Reviewer #1: Yes

Reviewer #2: Yes

2. Has the statistical analysis been performed appropriately and rigorously? 

Reviewer #1: Yes

Reviewer #2: Yes

3. Have the authors made all data underlying the findings in their manuscript fully available?

Reviewer #1: No

Reviewer #2: Yes

4. Is the manuscript presented in an intelligible fashion and written in standard English?

Reviewer #1: No

Reviewer #2: Yes

5. Review Comments to the Author

Reviewer #1: Title: Individual and contextual factors associated with abortion among reproductive age women in Sub-Saharan Africa: a generalized linear mixed effect modeling is indeed the very good evidence, but before reading to publication there many thing that need consideration

General comments

Please give special attention to the spelling and grammars as there some mechanical errors throughout the document. You may need editing by fluent speakers.

Please try to formulate your conclusion based on your findings and any potential interventions you may have made. Your conclusion seems to be justified.

Reduce undefined terms and statements throughout the document. There ideas which not reference and more personal.

You need EDHS in title with appropriate modification

Consequently you need to describe what sources o data you used

How you dealt missing values?

Appending datasets is a bit new for many scholars; I request more detailed method section so that lowers can learn from it to combine datasets.

discussion should include ‘strength and limitation’ header and add also ‘implication of study’

Abstract introduction should written positive statements that lead to negative that caused the gap then what this study could do. I would always suggest authors to follow the following way to write

Abstract

The abstract is good but confusing and does not leave any impressive impression. I suggest redrafting it according to the following guidelines.

The first two lines must encompass the context of the study and the research problem, further two lines must be covered the objective of the papers with unfolding the description of the title. In the next 2 to 4 lines the methodology will be covered. Afterward, the next two lines are for result and performance. In these lines, the author must define how the results and performance are being achieved, for instance, by conducting either simulation or physical implementation. Please mention the name of the simulation or the physical method. The result statistics must be mentioned in the last two lines and either in percentage or with real-time values.

Introduction:

The authors must give a clear procedure for the proposed solution with an algorithm (via flowcharts or pseudo codes, i.e. the flowcharts of a proposal work must be drawn), and must be supported with a figure either a block diagram of the proposed methodology or the topology in a formal style.

In the Literature Review section, the authors must cover the shortcomings because shortcoming or challenges in previous work is not mentioned adequately. If the previous work is free of challenges and there is no issue then what motivated the authors to propose this study? It inculcates that authors should revise the literature review and critically highlight the problems in the previous study and compare the proposed solution and tells how the proposed solution is best fitted.

In addition, please add a table and present the summary of related work, the shortcomings, and the proposed solution accordingly.

Method

In the Research Methodology section, what is the difference between your proposed method and other based techniques? (II) Please compare your proposed method with other based methods in terms of time complexity. Your method section has similarities (redundancy) with the previous studies on this dataset and which shows lack of novelty.

Critical analysis of the finding, which is the most important part, is missing and poorly addressed. This would help the readers to further improve the study.

Go for a thorough proofread of the paper to rectify several existing typos and grammatical mistakes to improve the written quality of the paper. If necessary, take the help of a native English speaker to improve the language of the whole paper.

Generally more detailed method section.

Discussion:

Rates of wasting in various countries is provided in both the introduction and the discussion. It is only needed in one of these. Providing this in the discussion for comparison to the present study findings would help to streamline the paper.

Conclusion:

Regarding the conclusion paragraph, Please precisely describe the outcome of the study and justify the statements that are mentioned in the abstract. Further, it must contain additional points and must give a clear and more discussion about the experimental results. The main novelty and contribution of needs must be summarized and highlight the recommendations based on obtained results. These results are the hallmark for future extension therefore, please spend some more time writing the conclusion and based on the results suggest new directions.

References

Please reduce outdated references and instead cite recent references that reflects your finding in context

Reviewer #2: abstarct:

1.The description of the model selection process could be expanded. For instance, mentioning how model fit was assessed (e.g., AIC, BIC) and why the model with the lowest deviance was chosen would be beneficial.

2.The results could be more detailed. For example, indicating the specific odds ratios for each factor and their significance levels would provide a clearer understanding of the strength of associations.

3.The conclusion could be strengthened by briefly discussing potential implications for policy and practice, as well as suggesting areas for future research to further address the issue.

Introduction:

1.The introduction could be more concise in some sections. For example, the sentence beginning with "Abortion can occur with medical intervention..." could be shortened or split into two for clarity.

2.The phrase "Even though one of the Sustainable Development Goal is to reduce global maternal mortality ratio less than 70 per 100,000 live births by 2030..." could be streamlined for readability.

3.There are a few grammatical issues that need correction. For example, "Even though, Abortion is undeniably associated..." should be "Even though abortion is undeniably associated..."

4.Ensure consistent use of capitalization, particularly with terms like "abortion," which is sometimes capitalized and sometimes not.

5.While the statistics are informative, providing some context or interpretation would make them more meaningful. For instance, when discussing the doubling of annual abortions in SSA, a brief mention of contributing factors (e.g., lack of access to contraception) would add depth.

6.The aim of the study is mentioned at the end, but it could be more specific. Instead of just stating that the study will investigate the pooled prevalence and determinants of abortion, consider briefly outlining the specific factors or hypotheses being tested.

Method:

1. While the operational definition of media exposure is provided, other variables, especially those that might be subject to interpretation (e.g., wealth index, distance from health facility), could also benefit from clear operational definitions.

2. The methods section does not mention how missing data were handled. Given the large dataset and the potential impact of missing data on the analysis, it would be important to address this. Was any imputation done, or were cases with missing data excluded?

3.The choice of a p-value < 0.2 for selecting variables in the bi-variable analysis is mentioned, but it would be beneficial to provide a brief justification for this threshold. This is higher than the conventional 0.05 threshold typically used for statistical significance, so a rationale (e.g., ensuring that potentially relevant variables are not excluded too early) would help the reader understand the decision.

4. The methods section does not explicitly address how potential confounding variables were handled. It would be useful to include a statement on whether any steps were taken to identify and control for confounders in the analysis.

5.The methods section does not mention whether interaction terms between individual-level and community-level variables were considered in the analysis. Interaction terms can provide insights into how the effects of individual-level factors might vary across different community contexts. If not considered, it might be worth discussing why, or if they were considered, how they were handled in the models.

Results:

1.The term "history of terminated pregnancy" is used to describe a variable, but it is not immediately clear whether this refers to both spontaneous and induced abortions or only to induced abortions. It would be beneficial to clarify this in the text to avoid any potential confusion.

2.The flow of the results could be improved by grouping related findings together more effectively. For example, discussing age-related findings together before moving on to marital status, education, and other variables would create a more cohesive narrative. This could also be achieved by organizing the findings into subsections.

3.The results mention that a large portion of the study population comes from Western Africa, with smaller portions from Eastern, Central, and Southern Africa. However, the results do not elaborate on whether there were significant differences in abortion prevalence across these regions. A more detailed analysis of regional differences could add depth to the findings.

4.Although the data is presented in detail, there is no mention of whether the observed differences are statistically significant. Including p-values or confidence intervals for key comparisons would provide insight into the robustness of the findings.

5.There is some inconsistency in reporting percentages, particularly when discussing the likelihood of abortion among different groups. For example, it is noted that rural residents have a 9% lower likelihood of abortion (AOR=0.91), but the same level of detail is not consistently applied across all variables.

6.The results for parity show a decrease in the likelihood of abortion as the number of births increases (e.g., grand multiparous women have lower odds of abortion compared to nulliparous women). This is an interesting finding

7.The section mentions the pooled prevalence of abortion across different regions of SSA, but there is little contextualization or discussion of why these regional differences might exist. Adding a brief analysis of regional socio-cultural or economic factors that could influence these differences would enhance the understanding of the findings.

8.The section includes model comparison statistics, such as deviance, AIC, and BIC, which are useful for determining the best-fit model. However, a brief explanation of what these statistics represent and why the final model was chosen would be beneficial for readers who may not be familiar with these concepts.

9. The tables provided are detailed and informative.

conclusion:

Instead of stating "The prevalence of abortion in SSA is high," consider "The study indicates a high prevalence of abortion in SSA, highlighting its significance as a public health issue."

6. PLOS authors have the option to publish the peer review history of their article (what does this mean?). If published, this will include your full peer review and any attached files.

Reviewer #1: **Yes: **Girma Gilano

Reviewer #2: No

---

## [Author Response · Author response to Decision Letter 0]

18 Sep 2024

Reviewer #1: Title: Individual and contextual factors associated with abortion among reproductive age women in Sub-Saharan Africa: a generalized linear mixed effect modeling is indeed the very good evidence, but before reading to publication there many thing that need consideration

Since the initial draft of the manuscript was written over a year ago, numerous updated DHS datasets have been published. Consequently, it was necessary to perform a reanalysis using the most recent datasets from 2015 to 2023. Therefore, we have conducted a reanalysis utilizing these recent datasets.

General comments

Please give special attention to the spelling and grammars as there some mechanical errors throughout the document. You may need editing by fluent speakers.

Author’s response: Dear reviewer thank you for your comment. We have thoroughly read and revised the manuscript and correct mechanical errors. 

Please try to formulate your conclusion based on your findings and any potential interventions you may have made. Your conclusion seems to be justified.

Author’s response: Dear reviewer thank you for your comment. We have revised the conclusion as per your suggestion. 

Reduce undefined terms and statements throughout the document. There ideas which not reference and more personal.

Author’s response: Dear reviewer thank you for your comment. We have thoroughly read and revised the manuscript and correct such errors. 

You need EDHS in title with appropriate modification

Author’s response: Dear reviewer thank you for your comment. we do not utilized Ethiopian demographic and health survey (EDHS) only rather we used DHS surveys from 24 SSA countries 

Consequently you need to describe what sources o data you used

Author’s response: Dear reviewer thank you for your comment. We stated that our source of data source were DHS survey and the dataset were granted after obtaining permission form the DHS program on the method section, ethical consideration and declaration parts of the manuscript. 

How you dealt missing values?

Author’s response: Dear reviewer thank you for your comment. we handled missing values in accordance with the DHS guidelines. 

Appending datasets is a bit new for many scholars; I request more detailed method section so that lowers can learn from it to combine datasets.

Author’s response: Dear reviewer thank you for your suggestion. We have included a statement regarding what appending is the STATA command as well under data management and analysis. 

discussion should include ‘strength and limitation’ header and add also ‘implication of study’

Author’s response: Dear reviewer thank you for your suggestion. We have added ‘strength and limitation’ and ‘implication of study’. 

Abstract introduction should written positive statements that lead to negative that caused the gap then what this study could do. I would always suggest authors to follow the following way to write

Abstract

The abstract is good but confusing and does not leave any impressive impression. I suggest redrafting it according to the following guidelines.

The first two lines must encompass the context of the study and the research problem, further two lines must be covered the objective of the papers with unfolding the description of the title. In the next 2 to 4 lines the methodology will be covered. Afterward, the next two lines are for result and performance. In these lines, the author must define how the results and performance are being achieved, for instance, by conducting either simulation or physical implementation. Please mention the name of the simulation or the physical method. The result statistics must be mentioned in the last two lines and either in percentage or with real-time values.

Author’s response: Dear reviewer thank you for your comment. We have revised the abstract as per your comment. (Please see the revised manuscript) 

Introduction:

The authors must give a clear procedure for the proposed solution with an algorithm (via flowcharts or pseudo codes, i.e. the flowcharts of a proposal work must be drawn), and must be supported with a figure either a block diagram of the proposed methodology or the topology in a formal style.

Author’s response: Dear reviewer thank you for your comment. Apologies, but we couldn't quite grasp the meaning behind this comment.

In the Literature Review section, the authors must cover the shortcomings because shortcoming or challenges in previous work is not mentioned adequately. If the previous work is free of challenges and there is no issue then what motivated the authors to propose this study? It inculcates that authors should revise the literature review and critically highlight the problems in the previous study and compare the proposed solution and tells how the proposed solution is best fitted.

In addition, please add a table and present the summary of related work, the shortcomings, and the proposed solution accordingly.

Author’s response: Dear reviewer thank you for your suggestion. Previous studies on abortion in SSA have often been limited in scope, focusing on specific regions or particular age groups, such as adolescents. This has resulted in a fragmented understanding of the issue, highlighting the need for more comprehensive research that encompasses a broader demographic and geographic range. This study aims to fill that gap by examining the pooled prevalence and determinants of abortion among women of reproductive age across 24 SSA countries using the most recent Demographic and Health Surveys. We have included this statement in the introduction. Please see the revised manuscript. 

Method

In the Research Methodology section, what is the difference between your proposed method and other based techniques? (II) Please compare your proposed method with other based methods in terms of time complexity. Your method section has similarities (redundancy) with the previous studies on this dataset and which shows lack of novelty.

Author’s response: Dear reviewer thank you for your comment. While the methods section may resemble those of previous studies that utilized similar data sources, this does not imply plagiarism. We employed a multilevel analysis, which is the most suitable model for hierarchical datasets like DHS. This approach has also been used in prior research.

Critical analysis of the finding, which is the most important part, is missing and poorly addressed. This would help the readers to further improve the study.

Author’s response: Dear reviewer thank you for your comment. We would love to know what analysis is missed and not addressed so we can improve the quality of the manuscript. 

Go for a thorough proofread of the paper to rectify several existing typos and grammatical mistakes to improve the written quality of the paper. If necessary, take the help of a native English speaker to improve the language of the whole paper.

Author’s response: Dear reviewer thank you for your comment. We have proofread and rectify several existing typos and grammatical mistakes as per your comment. 

Generally more detailed method section.

Discussion:

Rates of wasting in various countries is provided in both the introduction and the discussion. It is only needed in one of these. Providing this in the discussion for comparison to the present study findings would help to streamline the paper.

Author’s response: Dear reviewer thank you for your comment. The rates on the discussion and introduction part are not similar. The rates on the introduction are abortion rates and maternal mortality. The rates mentioned in the discussion part are pregnancy rates and unwanted pregnancy rates which used as a potential explanation for the observed prevalence of abortion in SSA. 

Conclusion:

Regarding the conclusion paragraph, Please precisely describe the outcome of the study and justify the statements that are mentioned in the abstract. Further, it must contain additional points and must give a clear and more discussion about the experimental results. The main novelty and contribution of needs must be summarized and highlight the recommendations based on obtained results. These results are the hallmark for future extension therefore, please spend some more time writing the conclusion and based on the results suggest new directions.

Author’s response: Dear reviewer thank you for your comment. We have revised the conclusion as per your comments and suggestion. (Please see the revised manuscript). 

References

Please reduce outdated references and instead cite recent references that reflects your finding in context

Author’s response: Dear reviewer thank you for your comment. We have revised our references. 

Reviewer 2

abstarct:

1.The description of the model selection process could be expanded. For instance, mentioning how model fit was assessed (e.g., AIC, BIC) and why the model with the lowest deviance was chosen would be beneficial.

Author’s response: Dear reviewer thank you for your comment. Deviance is often preferred over AIC and BIC for assessing the goodness-of-fit in nested models because it allows for a direct comparison using the likelihood ratio test, which provides a p-value to determine statistical significance. This focus on model fit, without the complexity penalties inherent in AIC and BIC, makes deviance simpler and more interpretable for nested models. While AIC and BIC are useful for model selection, especially with non-nested models, deviance offers a clearer measure of improvement in fit between nested models. We have included this explanation in the Data management and analysis. (Please see the revised manuscript).

2.The results could be more detailed. For example, indicating the specific odds ratios for each factor and their significance levels would provide a clearer understanding of the strength of associations.

Author’s response: Dear reviewer thank you for your comment. We have added the specific adjusted odds ratios with their respective confidence intervals. 

3.The conclusion could be strengthened by briefly discussing potential implications for policy and practice, as well as suggesting areas for future research to further address the issue.

Author’s response: Dear reviewer thank you for your comment. We have indicated potential implications of the finding and suggest areas for future research and intervention. 

Introduction:

1.The introduction could be more concise in some sections. For example, the sentence beginning with "Abortion can occur with medical intervention..." could be shortened or split into two for clarity.

Author’s response: Dear reviewer thank you for your comment. We have rewritten the sentence. 

2.The phrase "Even though one of the Sustainable Development Goal is to reduce global maternal mortality ratio less than 70 per 100,000 live births by 2030..." could be streamlined for readability.

Author’s response: Dear reviewer thank you for your comment. We have revised the sentence. 

3.There are a few grammatical issues that need correction. For example, "Even though, Abortion is undeniably associated..." should be "Even though abortion is undeniably associated..."

Author’s response: Dear reviewer thank you for your comment. We have corrected the grammatical error. 

4. Ensure consistent use of capitalization, particularly with terms like "abortion," which is sometimes capitalized and sometimes not.

Author’s response: Dear reviewer thank you for your comment. We have corrected the grammatical error. 

5. While the statistics are informative, providing some context or interpretation would make them more meaningful. For instance, when discussing the doubling of annual abortions in SSA, a brief mention of contributing factors (e.g., lack of access to contraception) would add depth.

Author’s response: Dear reviewer thank you for your comment. We have the potential contributing factors. Please see the revised manuscript

6. The aim of the study is mentioned at the end, but it could be more specific. Instead of just stating that the study will investigate the pooled prevalence and determinants of abortion, consider briefly outlining the specific factors or hypotheses being tested.

Author’s response: Dear reviewer thank you for your comment. We have outlined the hypothesis we aim to test at the end of the introduction section.

Method:

1. While the operational definition of media exposure is provided, other variables, especially those that might be subject to interpretation (e.g., wealth index, distance from health facility), could also benefit from clear operational definitions.

Author’s response: Dear reviewer thank you for your suggestion. We have provided operational definition for both distance to health facility and wealth index. 

2. The methods section does not mention how missing data were handled. Given the large dataset and the potential impact of missing data on the analysis, it would be important to address this. Was any imputation done, or were cases with missing data excluded?

Author’s response: Dear reviewer thank you for your comment. We handled missing values in accordance with the DHS guidelines.

3. The choice of a p-value < 0.2 for selecting variables in the bi-variable analysis is mentioned, but it would be beneficial to provide a brief justification for this threshold. This is higher than the conventional 0.05 threshold typically used for statistical significance, so a rationale (e.g., ensuring that potentially relevant variables are not excluded too early) would help the reader understand the decision.

Author’s response: Dear reviewer thank you for your suggestion. As you have mentioned the choice of a p-value < 0.2 for selecting variables in the bi-variable analysis is intentional to ensure that potentially relevant variables are not excluded too early in the analysis process. This threshold is higher than the conventional 0.05 used for statistical significance, allowing us to capture a broader range of variables that may have an impact. By doing so, we aim to include variables that might show significance in the multivariable analysis, thus providing a more comprehensive understanding of the factors at play. We have included this statement in the method section. 

4. The methods section does not explicitly address how potential confounding variables were handled. It would be useful to include a statement on whether any steps were taken to identify and control for confounders in the analysis.

Author’s response: Dear reviewer thank you for your suggestion. In this study, we took several steps to identify and control for potential confounding variables. Initially, we conducted a thorough literature review to identify common confounders associated with our primary variables of interest. We then used statistical methods such as multivariable regression analysis to adjust for these confounders.

5.The methods section does not mention whether interaction terms between individual-level and community-level variables were considered in the analysis. Interaction terms can provide insights into how the effects of individual-level factors might vary across different community contexts. If not considered, it might be worth discussing why, or if they were considered, how they were handled in the models.

Author’s response: Dear reviewer thank you for your question. We do not consider interaction term in the analysis. Including interaction terms can significantly increase the complexity of the statistical models. We opted for a simpler model to ensure clarity and interpretability of the results. Furthermore, the primary focus of the study might have been on the main effects of individual-level and community-level variables, rather than their interactions. We have prioritized understanding these main effects first. 

Results:

1.The term "history of terminated pregnancy" is used to describe a variable, but it is not immediately clear whether this refers to both spontaneous and induced abortions or only to induced abortions. It would be beneficial to clarify this in the text to avoid any potential confusion. 

Author’s response: Dear reviewer thank you for your question. The DHS does not distinguish between spontaneous and induced abortion. We have acknowledged this limitation under strength and limitation 

---

## [Editor Report · Decision Letter 1]

25 Nov 2024

Individual and contextual factors associated with abortion among reproductive age women in Sub-Saharan Africa: insights from 24 recent demographic and health surveys

PONE-D-23-16508R1

Dear Dr. Seifu,

We’re pleased to inform you that your manuscript has been judged scientifically suitable for publication and will be formally accepted for publication once it meets all outstanding technical requirements.

Kind regards,

Alfredo Luis Fort, M.D., M.Sc., Ph.D.

Academic Editor

PLOS ONE

Additional Editor Comments (optional):

Thank you for addressing all the aspects highlighted by the previous reviewers. You have provided a more substantive and well-presented manuscript. However, there are several additional edits required for the benefit of the readers. Please see them in the attached file, to correct before submitting the final manuscript. Thank you.

---

## [Editor Report · Acceptance letter]

2 Dec 2024

PONE-D-23-16508R1 

PLOS ONE

Dear Dr. Seifu, 

I'm pleased to inform you that your manuscript has been deemed suitable for publication in PLOS ONE. Congratulations! Your manuscript is now being handed over to our production team.

Kind regards, 

on behalf of

Dr. Alfredo Luis Fort 

Academic Editor

PLOS ONE